# Analysis of Fifty Years of Severe Malaria Worldwide Research

**DOI:** 10.3390/pathogens12030373

**Published:** 2023-02-24

**Authors:** Jose A. Garrido-Cardenas, Lilia González-Cerón, Federico García-Maroto, José Cebrián-Carmona, Francisco Manzano-Agugliaro, Concepción M. Mesa-Valle

**Affiliations:** 1Department of Biology and Geology, University of Almeria, 04120 Almeria, Spain; 2Regional Center for Public Health Research, National Institute of Public Health, Tapachula 30700, Chiapas, Mexico; 3Department of Chemistry and Physics, University of Almeria, 04120 Almeria, Spain; 4Department of Engineering, University of Almeria, 04120 Almeria, Spain

**Keywords:** plasmodium falciparum, malaria pathogenesis, malaria treatment, severe malaria symptoms, malaria epidemiology

## Abstract

This study analyzed fifty years of severe malaria research worldwide. Malaria is a parasitic disease that continues to have a significant impact on global health, particularly in sub-Saharan Africa. Severe malaria, a severe and often fatal form of the disease, is a major public health concern. The study used different bibliometric indicators such as the number of publications, citations, authorship, and keywords to analyze the research trends, patterns, and progress made in the field of severe malaria. The study covers the period from 1974 to 2021 and includes articles from Scopus. The results of the study indicated that there has been a steady increase in the number of publications on severe malaria over the past fifty years, with a particular increase in the last decade. The study also showed that most of the publications are from USA and Europe, while the disease occurs in Africa, South-East Asia, and the Americas. The study also identified the most frequent keywords used in the publications, and the most influential journals and authors in the field. In conclusion, this bibliometric study provides a comprehensive overview of the research trends and patterns in the field of severe malaria over the past fifty years and highlights the areas that need more attention and research efforts.

## 1. Introduction

Malaria is one of the deadliest infectious diseases in the world. It is caused, in humans, by six species of parasitic protozoa of the genus *Plasmodium*: *P. falciparum* (responsible for most deaths, and mainly present in Africa), *P. vivax* (the species with the largest geographical distribution outside Africa), *P. knowlesi*, *P. malariae*, *P. ovale wallikeri*, and *P. ovale curtisi* [1]. Most reported cases of severe malaria are caused by *P. falciparum*, although the burden of *P. vivax* and *P. knowlesi* cannot be neglected [2].

The term severe malaria was first described in 1985 at the first WHO meeting convened by the Malaria Action Programme of the World Health Organization, and since then, this complication of the disease has been the focus of global research [3].

Mortality due to complications of severe malaria depends on multiple factors, such as the medical circumstances surrounding the patient, the treatment received, and the course of the infection itself. Patient frailty due to factors not directly associated with malaria, such as age, co-infection with viruses or bacteria, or a weak immune system, can lead to death, regardless of the infecting parasite [4,5]. Severe malaria usually has a mortality rate of more than 5%, which is high compared to uncomplicated malaria cases, where the mortality rate is as low as 0.1% [6]. Severe malaria is generally evenly distributed across all population groups in areas of low malaria transmission, but the situation is different in areas of high transmission, where the risk is higher among non-immune people such as young children and visitors from non-endemic areas [7,8].

Although severe malaria is often defined in epidemiological or research terms, the most practical way of delineating it is usually associated with the clinical features or the vital organ dysfunction it causes. Thus, the most prevalent clinical features in patients with severe malaria are: generalized weakness that may lead to the prostration of the patient; continued failure of consciousness and other neurological abnormalities; profound anemia, with the presence of hemoglobin in the urine (also due to acute kidney injury); respiratory problems, with increased frequency and frequent pauses; circulatory collapse or shock, with systolic blood pressure < 80 mmHg in adults and <50 mmHg in children; hemorrhages; pulmonary edema; convulsions; and clinical jaundice plus evidence of other vital organ dysfunction [9]. These clinical manifestations may occur in isolation or, more frequently, in combination in the same patient [10]. In most cases, these pathological events are associated with the sequestration of erythrocytes containing metabolically active parasites, which causes alterations in the coagulation and cytopreservation pathways [11].

According to WHO estimates, the majority of malaria cases do not develop into severe malaria. This only occurs in less than 1% of diagnosed cases (619,000 deaths out of 247 million reported cases, according to the most recent data) [12]. Of the hundreds of thousands of deaths each year, two-thirds involve children under five years. Children hospitalized with anemia resulting from an episode of severe malaria have a high mortality rate in the months following admission [13].

Even in children who survive the disease, the consequences are often dire. In about 10% of cases, a neurological deficit occurs causing seizures after recovery, in addition to a clinical picture of stroke. It is suggested that this is because a large vascular territory of the brain area is involved in the development of the disease. In addition, other mental and behavioral problems often appear with increasing frequency [14,15].

It is, therefore, necessary to continue working to improve early diagnosis and treatment in cases of severe malaria, as well as to prevent the after-effects of the disease. To this end, it is essential to know how far we have come in the last five decades and where we are now.

## 2. Materials and Methods

The methodology for this bibliometric study on the analysis of fifty years of severe malaria worldwide research will involve the following steps:

Data collection: The study used a comprehensive search strategy to collect data from the Scopus database. The choice of the Scopus database was based on the idea that it has the most articles, journals, books, and publishers indexed [16]. Scopus is the database developed by Elsevier, with publications from 1788 to the present day, and its use offered a sample size that is statistically sufficiently representative of what is to be shown. For this reason, Scopus is the most widely used database in bibliometric analyses of areas as different as medicine [17], social sciences [18], or agriculture [19]. In this case, the search was limited to articles published between 1974 and 2021, in English and other languages, such as Spanish, French, Russian, Chinese, etc. The search terms included keywords related to severe malaria such as “severe malaria”, “malaria pathogenesis”, “malaria treatment” and “malaria epidemiology”.

Data screening: The collected data were screened to ensure that they met the inclusion criteria, which included articles on severe malaria and its management. Articles that did not meet the inclusion criteria were excluded from the study.

Data analysis: The data was analyzed using bibliometric indicators such as the number of publications, citations, authorship patterns, and keywords. The data were analyzed using the software VOSviewer (version 1.6.18).

Results and discussion: The results are presented in tables and figures to show the research trends and patterns in the field of severe malaria over the past fifty years. The results were also discussed in the context of the existing literature on causes or complications leading to severe malaria.

Conclusion: The study highlighted the areas that need more attention and research efforts in the field of severe malaria and the implications of the findings for future research in this field.

## 3. Results

### 3.1. Global Evolution Trend of Scientific Output

The search returned 3794 documents. Figure 1 shows the global evolution trend of the number of documents on severe malaria since the first article was published, from 1974 until 2021, which is the year with all updated data. As can be observed in Figure 1, these five decades were divided into three stages. The first of these runs from 1974 to 1986. In this period, 25 articles were published, with no trend of growth over time. The real take-off in publications on severe malaria occurred from 1987 onwards, and a linear growth trend can be observed until 2009, which was only altered in 2001 when there was a relative low that broke the trend. The third stage covers the period from 2009 onwards, and it is in this stage that the highest number of articles published per year in the entire chronology studied was recorded. Specifically, this last stage showed an average of 184 articles published each year.

Since the late 1980s, severe malaria has received increasing attention from the international scientific community (Figure 1). From then until now, the disease has been very present in the scientific literature. In two of the years (2012 and 2014), relative peaks were reached with more than 200 articles published.

### 3.2. Publication Distribution by Authors, Institutions, and Countries

The 12 most important authors in severe malaria with at least fifty publications each are shown in Table 1. The percentage of publications on severe malaria ranged from 8.8% (Nicholas P.J. Day, of the Mahidol Oxford Tropical Medicine Research Unit, in Bangkok, Thailand) to 30.2% (Robert Opika Opoka, of Makerere University, in Kampala, Uganda).

The 12 authors belonged to institutions from 7 different countries (Thailand, Kenya, Germany, Australia, Canada, United Kingdom, and Uganda), and above all, the researcher Nicholas John White, from the Faculty of Tropical Medicine, in the Mahidol University (Bangkok, Thailand), stood out with 169 publications on severe malaria. Nicholas J. White is a British medical researcher who has devoted his research to tropical medicine, especially malaria, in developing countries. White has established important research networks throughout his research life and has co-written articles with more than 3000 researchers from all over the world, some of them presented in Table 1, such as AM Dondorp or NPJ Day.

Figure 2 shows the 10 institutions with at least 100 publications on severe malaria. As can be noted, some of the institutions listed in Table 1 also appear in this list, but not all of those in Figure 2 appear in Table 1. This is the case of the University of Oxford, which appears first in the ranking, with almost 10% of the publications on severe malaria, but does not appear in Table 1.

In Figure 3, the countries with the most publications on severe malaria are shown. Among them, the United Kingdom and the United States stand out, with 932 and 808 articles, respectively. Following them are six countries with between 250 and 500 publications. These are Thailand (360), France (340), Kenya (308), India (294), Australia (276), and Germany (264). Finally, 13 countries are shown publishing between 100 and 250 articles.

These 21 countries are also listed in Table 2. In this Table, not only the total number of articles published in each country has been taken into account, but also the population and GDP per capita have been considered. The total population data were obtained from https://www.worldometers.info/world-population/population-by-country/ (accessed on 1 October 2022), while GDP per capita data, from https://www.worldometers.info/gdp/gdp-per-capita/ (accessed on 1 October 2022). GDP (Gross Domestic Product) per capita is shown in dollars and represents a country’s GDP divided by its total population.

As mentioned above, in absolute terms, it can be observed that among the 21 countries that have published at least 100 articles on this subject, there are two that stand out above the rest, the United Kingdom and the United States. However, if we consider the number of inhabitants of each country and their relative wealth, measured in terms of GDP, we can draw conclusions that allow us to establish a real profile of research in this area.

Firstly, of all the countries analyzed, only 7 have a P/N value (number of publications per million inhabitants) higher than 10. These are, in increasing order of P/N: Australia, Sweden, United Kingdom, Denmark, Switzerland, and, above all, Gambia and Gabon. These data can be seen in Figure 4. In Figure 4, the P/N data have been plotted against the per capita income of each country. It can be seen how the most developed countries—those with a GDP per capita above 40,000—are shown at the top of the graph, and the low and middle-income countries—those with a GDP per capita below 20,000—are shown at the bottom of the graph. However, in both groups of countries, two subgroups appear on the right, representing those countries with higher P/N values, with Denmark and Switzerland standing out among the more developed countries, and, above all, Gambia and Gabon among the less developed ones.

### 3.3. Keyword Analysis and Drugs

When analyzing the 160 keywords with the highest presence in the articles published on severe malaria, it was found that the most important element that most articles on severe malaria focus on are drugs and that there are 8 drugs used in the treatment of the disease in the keywords. These drugs are: Artemisinin derivative (appearing as a keyword in 828 articles), Quinine (in 739 articles), Chloroquine (in 381 articles), Mefloquine (in 235 articles), Artemether Plus Benflumetol (in 204 articles), Doxycycline (in 178 articles), Fansidar (in 145 articles), and Primaquine (in 132 articles). Figure 5 shows the evolution over time of the appearance of these keywords in the scientific literature.

The figure shows three-time intervals, very similar to those shown in Figure 1, with different trends. First, there is a period from 1970 to 1990. In this period, virtually no drugs appeared among the keywords in severe malaria articles, and only Quinine, Chloroquine, and Mefloquine appeared in at least 10 articles. The other two periods share almost 50% of the remaining drug occurrences among the keywords. These are the periods 1991–2009 and 2010–2021. However, not all drugs were evenly distributed over the years in both periods. Quinine, Chloroquine, Mefloquine, and Fansidar appeared mostly in the 1991–2009 period, while Artemisinin Derivative, Artemether Plus Benflumetol, and Primaquine appeared mostly in the more recent period. Interestingly, it was noted that artemisinin derivatives appeared in more than 80% of the articles in which a drug appears, while Fansidar has not appeared in the scientific literature analyzed since 2012.

## 4. Discussion and Conclusions

The evolution of the number of publications on severe malaria over the last fifty years was studied. The first thing to note is that, since the late 1980s, severe malaria has received increasing attention from the international scientific community. This can be partly explained by the fact that the World Health Organization hosted an informal technical meeting on severe and complicated malaria, in June 1985. Before this date, there was no standard definition of severe malaria. The outcome of that first meeting was a review of the existing knowledge on severe malaria and the establishment of guidelines for clinical case management aimed at reducing mortality as far as possible at all levels of health services. Three years later, in March 1988, a second meeting was held, which allowed the previously established measures to be highlighted and updated. As a result, in 1990, the World Health Organization (WHO) established a series of criteria to assist future clinical and epidemiological studies in order to optimize resources in the fight against this form of the disease [20]. Subsequently, the third and fourth WHO meetings on severe malaria were held in 1995 and 2013, at which malaria issues continued to be addressed. In addition, modifications to the general definition of severe malaria were proposed to include severe disease caused by *Plasmodium vivax* and *Plasmodium knowlesi* infections [2].

When analyzing the publication distribution by authors, institutions, and countries, several conclusions could be drawn. The first of these has to do with the absence of major institutions, such as the University of Oxford, in the list of main authors and their affiliation. In the specific case of Oxford University, we could find two possible justifications for this fact. On one hand, researchers from this University have multiple collaborations with some of the most important researchers in this area, such as NJ White or AM Dondorp; on the other hand, at the University of Oxford, there are researchers, such as Dominic P. Kwiatkowski, with a high number of publications, but that not appear in Table 1, as they do not have 50 publications on severe malaria. It is likely that for other institutions such as the Nuffield Department of Medicine (at the University of Oxford) or the Kenya medical research institute, this argument is also valid.

Regarding the data obtained on the countries in which most articles are published on severe malaria, it can be observed that when these data are considered together with those of population and GDP per capita, it reveals the type of relationship that has been established between what we could call poor countries and rich countries, in strategic alliances in the fight against the disease. In any case, several reasons would justify the position of the top countries in this ranking. As Figure 4 shows, four countries stand out from the rest: Denmark, Switzerland, Gabon, and Gambia.

Denmark’s research is supported by the existence of the Centre for Medical Parasitology (CMP), which focused on malaria. It was founded in 1991, due to the collaboration between research groups at the Institute for Medical Microbiology and Immunology, the Institute of Public Health, University of Copenhagen, and the Departments of Infectious Diseases and Clinical Microbiology at Rigshospitalet. The main current research topics of the CMP are drug resistance, parasite biology and immunology, vaccine development, and intervention studies in high-risk populations, all of which are related to severe malaria. The presence of Switzerland could be explained by the well-established pharmaceutical industry in the country and the existence of numerous research and development institutions, both public and private, that revolve around this type of industry. In addition, Switzerland is home to international organizations and institutions responsible for global health care, such as the Swiss Agency for Development and Cooperation (SDC) and the World Health Organization, which is headquartered in Geneva.

As far as low- and middle-income countries are concerned, the justification for finding Gabon in such a prominent position is probably due to the ability of the country’s situation and its academics to attract international collaborations. Among them, it is worth highlighting the presence of Peter Kremsner, who appears in 80 of the 107 articles published in Gabon. Kremsner is a highly cited scientist in the field of malaria, who directs the Centre de Recherches Médicales de Lambaréné (CERMEL), Gabon. He is a specialist in tropical medicine and a full professor at the University of Tübingen, Germany. Kremsner and his research team have developed a simple method to assess the severity of malaria by analyzing coma and deep breathing, reducing the need for laboratory assessment.

As for The Gambia, its role can be justified by the presence of the MRC Unit The Gambia at LSHTM (London School of Hygiene and Tropical Medicine). This unit is one of two established in sub-Saharan Africa by the UK Medical Research Council, which financially supports research in the country. The MRC Unit The Gambia represents an extraordinary concentration of scientific expertise and research platforms, which are unique in West Africa. This enables the development of scientific partnerships between UK research centers such as the University of Oxford, the London School of Hygiene & Tropical Medicine, and the Wellcome Centre for Human Genetics, with Gambian researchers, so that both basic and applied research into the control of diseases such as malaria are encouraged.

Concerning the data obtained from the analysis of the drugs among the keywords in the articles on severe malaria, we can see how their appearance over the years gives us clues about the WHO recommendations at any given time, as well as the state of research in each period. The first thing that emerges is that two types of drugs are at the forefront of research worldwide. These are the artemisinin derivatives and quinine [21]. Internationally, artemisinin derivatives have been the treatment of choice since their intravenous administration was shown to be more efficacious than intravenous quinine in both pediatric and adult patients [22]. It is better at both reducing mortality and clearing blood parasites in patients of all ages. For a fairly long period, quinine was the standard treatment used for the treatment of severe malaria, but its use has been associated with episodes of resistance, as well as adverse effects such as deafness and hypoglycemia. For this reason, since 2011, artemisinin derivatives have been the preferred antimalarials for treating severe *P. falciparum* malaria, as they have been shown to kill the parasites more quickly than quinine [23]. However, these treatments are not foolproof and artemisinin resistance has been reported, which calls for further therapies and better use of the current drugs.

In conclusion, although advances in our understanding of severe malaria are very important, more research is needed in this area as much remains to be done. There are no definitive treatments to prevent malaria from progressing to severe malaria, nor is it possible to prevent the sequelae of the disease. Similarly, it is of utmost importance to deepen our knowledge of the development of severe malaria caused by *P. vivax* and *P. knowlesi*. The mortality caused by the disease, despite the improvement in recent years, is still more than worrying. Therefore, it is necessary to review current strategies for both treatment and prevention of the disease. To this end, the alliances established between research groups around the world are the only tool that can open new avenues in the world of research. This study also highlighted the need for more research efforts in areas such as the epidemiology, pathogenesis, and management of severe malaria, in order to develop more effective interventions and reduce the burden of this disease worldwide.

## Figures and Tables

**Figure 1 pathogens-12-00373-f001:**
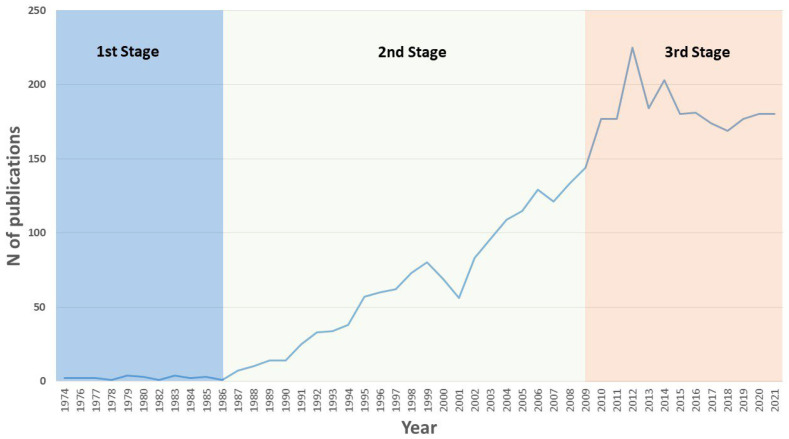
Trend in the number of publications per year on severe malaria over the last fifty years.

**Figure 2 pathogens-12-00373-f002:**
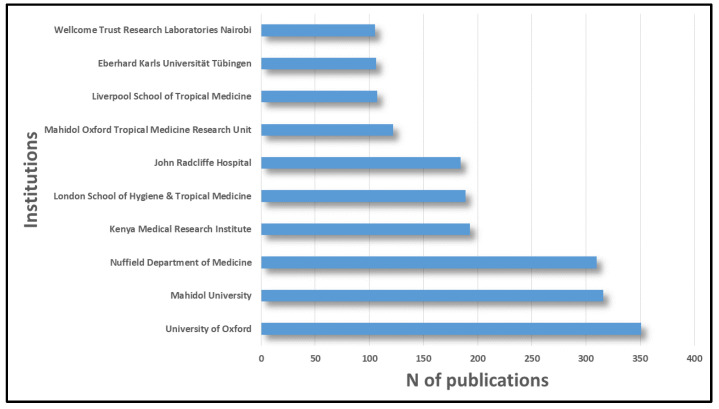
Top institutions whose researchers published the most articles on severe malaria.

**Figure 3 pathogens-12-00373-f003:**
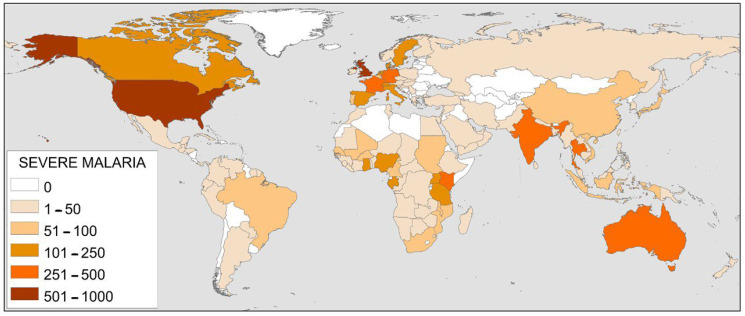
World map with the countries publishing the most articles on severe malaria.

**Figure 4 pathogens-12-00373-f004:**
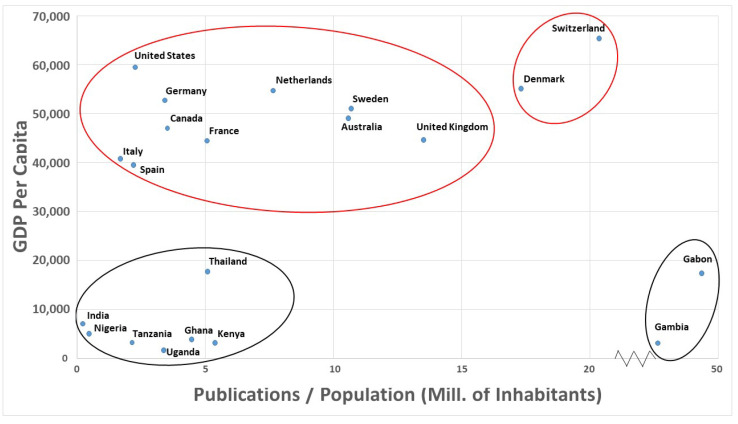
Representation of P/N (number of publications per million inhabitants) versus per capita income, for each country. Countries with a GDP per capita above 40,000 have been clustered with red circles, while countries with a GDP per capita below 20,000, with black circles.

**Figure 5 pathogens-12-00373-f005:**
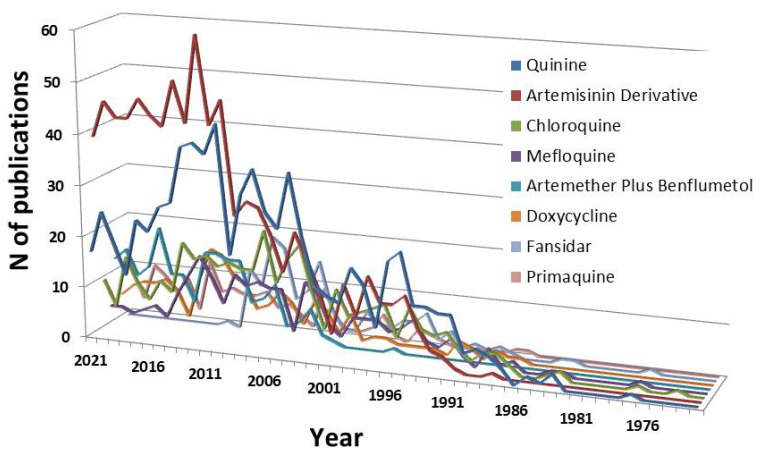
Evolution over time of the occurrence of drugs in the keywords in the articles analyzed on severe malaria.

**Table 1 pathogens-12-00373-t001:** Main authors highlighting severe malaria, the institutions to which they belong, and the countries in which these institutions are located.

Author	N	P	N%	Institution	Country
White, N.J.	169	1296	13.0	Mahidol University	Thailand
Marsh, K.	114	520	21.9	Centre for Geographic, Medicine Research	Kenya
Kremsner, P.G.	90	751	12.0	Eberhard Karls Universität, Tübingen	Germany
Dondorp, A.M.	78	526	14.8	Mahidol University	Thailand
Looareesuwan, S.	75	512	14.6	Hospital for Tropical Diseases	Thailand
Day, N.P.J.	67	785	8.5	Mahidol Oxford Tropical, Medicine Research Unit	Thailand
Anstey, N.M.	65	346	18.8	Menzies School of Health, Research	Australia
Kain, K.C.	60	401	15.0	University Health Network, University of Toronto	Canada
Krishna, S.	59	281	21.0	University of London	United Kingdom
Newton, C.R.J.C.	56	501	11.2	Pwani University	Kenya
Maitland, K.	54	201	26.9	Imperial College London	United Kingdom
Opoka, R.O.	51	169	30.2	Makerere University	Uganda

N: Number of severe malaria publications. P: Number of total publications.

**Table 2 pathogens-12-00373-t002:** Countries publishing the most on severe malaria, in terms of the total population and relative wealth.

Country	Publications (N)	Population (P) (Mill. of Inhabitants)	N/P	GDP Per Capita
United Kingdom	932	67.89	13.73	44,920
United States	808	331.00	2.44	59,928
Thailand	360	69.80	5.16	17,910
France	340	65.27	5.21	44,033
Kenya	308	53.77	5.73	3292
India	294	1410.75	0.21	7166
Australia	276	25.50	10.82	49,378
Germany	264	83.78	3.15	52,556
Switzerland	175	8.65	20.23	66,307
Uganda	173	45.74	3.78	1868
Nigeria	168	206.14	0.81	5887
Ghana	150	31.07	4.83	4502
Tanzania	148	59.73	2.48	2948
Netherlands	132	17.13	7.71	54,422
Canada	127	37.74	3.37	46,510
Italy	118	60.46	1.95	40,924
Sweden	110	10.10	10.89	51,405
Spain	108	46.75	2.31	39,037
Gabon	107	2.23	47.98	18,113
Gambia	103	2.42	42.56	1699
Denmark	102	5.79	17.62	54,356

## Data Availability

The data presented in this study are available in the article.

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
