# Peer review of "Analysis of Fifty Years of Severe Malaria Worldwide Research"

_pathogens, 2023, doi:10.3390/pathogens12030373_

Round 1
Reviewer 1 Report
The manuscript touches upon the very important topic of severe malaria and research progress on the topic in the last 50 years. It provides an important outlook on areas where the field has progressed, significant contributors, institutions, and economic correlations. It signifies why more work is important in the field for better intervention and treatment for malaria and how the affected areas can be better equipped with research.
Author Response
Thank you very much for your analysis. We are very pleased that you value the article and the work carried out.
Reviewer 2 Report
This study from Garrido-Cardenas and colleagues made an extensive bibliographic search on articles focused on severe malaria research. The article was easy to read and it was interesting to see a bibliometric analysis of severe malaria from a different perspective. However, I would recommend that authors review the phrasing of the full manuscript, but especially in a few specific sentences highlighted below. My overall suggestion is that authors are more specific with the language. Other interesting analyses could have done by the authors and suggestions are specified below, but this does not prevent publication of the article with the current analysis.
MAJOR:
Since the authors have performed such an extensive review, I would suggest to use more recent literature in the referenced articles.
Paragraph 49-60: I would suggest to cite the WHO severe malaria criteria found in “Management of Severe Malaria” from the WHO.
Results line 114-116: The authors should be more specific on the language or the time periods they referred to. For examples when they mention 184 articles/year. Is this in the 3rd stage? Or in the period between 2001 and 2021? Similarly, in line 137-139 the authors state that there were “more than 200 published over several years”, but according to the graph this only occurred in 2012 and 2014.
Line 163-165 and later in the discussion: “The case of the University of Oxford, which 163 appears first in the ranking, with almost 10% of the publications on severe malaria, but 164 does not appear in Table 1, is striking”. I wouldn’t classify it as striking. Kevin Marsh has dual affiliation at University of Oxford (Nuffield Department of Medicine, University of Oxford, Oxford, UK), and besides the University of Oxford has a strong research program on Malaria as the authors state in the discussion. I would suggest that the authors carefully look for dual affiliations of all the top listed authors.
Line 238-248: Comparison to GDP. Another important piece of information and interesting comparison that the authors are missing is a comparison with the % of the GDP dedicated to research.
Results: An important missing topic that is not covered in the text are articles that describe the physiopathology of the disease. Did the authors capture these with the 160 keywords used?
MINOR
Abstract: Line 20: Mentioning Nigeria, Ghana, Kenya specifically restricts malaria to Africa, and neglects the burden in other regions. Substitute for “in the tropics” or alternatively, “in Africa, South-East Asia and the Americas” following WHO regions
Intro line 34-35: Suggestion to substitute “Most reported cases of severe malaria are caused by P. falciparum, although the burden of P. vivax and P. knowlesi cannot be neglected.”
Line 45-48: Missing reference
Results:
Lines 142-143: “In column N shows the number of published publications on severe 142 malaria, while in column P shows, the number of total publications by each author.” This belongs to the table legend
Line 150: Missing word. 169 publications?
Line 198: Should refer to Table 2? Or Figure 3, which is not referenced in the text.
Line 202: Check grammar “Finally, 13 countries appear with between 100 and 250 articles.”
Line 273: The terms are drugs, not related to drugs “it is found that there are 8 terms related to drugs used in the treatment of the disease”. I would rephrase the sentence to highlight that most of severe malaria articles focus on the use of antimalarial drugs.
Line 354: Maybe rephrase “we can perfectly understand t” to “it reveals”
Author Response
Thank you very much. Your comments certainly improve the work. They have all been applied to the manuscript and are commented on in the attached document.
